# Deep Reinforcement Learning For Nash Equilibria in Non-Renewable Resource Differential Games

## Abstract

Characterizing Nash equilibria in oligopolistic non-renewable resource markets poses major challenges for computational economics, as traditional iterative methods face scalability limitations due to the curse of dimensionality. In this work, we propose a reinforcement learning–based approach to compute these equilibria and benchmark it against a modified iterative baseline, derived from an established algorithm for differential games and adapted to the oligopoly case. We conduct experiments in monopoly, duopoly, and multi-player settings, evaluating both reward accuracy and computational efficiency. Our results show that while iterative schemes provide good accuracy in low-dimensional problems, reinforcement learning scales more effectively to three- and four-player games, leading to a substantial reduction in computation time. This highlights the potential of reinforcement learning as a scalable tool for solving complex differential games in resource economics.

## 1 Introduction

Differential games are a powerful framework for modeling strategic interactions in economics, finance, and resource management. They capture settings in which multiple agents control state variables while interacting in shared market environments, yet their analysis is often constrained by high dimensionality and complex strategic interdependencies (6).

A particularly relevant class is that of non-renewable resource games, which pose fundamental challenges related to intergenerational equity, optimal extraction paths, and long-term environmental impacts. The energy transition has also emphasized the strategic importance of critical minerals, whose reserves are geographically concentrated and controlled by actors with significant market power (1; 2; 3; 4; 5). Consequently, reliable economic modeling of such settings must account for both the intertemporal nature of resource depletion and the strategic behavior of key market players.

The analysis of non-renewable resource games focuses on computing Nash equilibria, where no player can improve their payoff by unilaterally deviating from its strategy. In this paper, we focus on feedback (closed-loop) equilibria–state–dependent strategies that, while more computationally demanding, more faithfully capture the strategic adaptability of competitive resource markets (6).

Such games can be classified along several structural dimensions (Figure 11 in the Appendix A): whether strategies depend on stock levels (6) or only on time (14), the degree of information available to players, the form of the demand function (15), and the presence or absence of extraction costs (6). In this study, we focus on the case of linear demand without price caps, with full information on all stock levels, and adjust the cost structure according to the objectives of our analysis.

Characterizing equilibria in such markets requires solving coupled control problems whose dimensionality grows with the number of players and state variables (16). Traditional iterative approaches address this by strengthening the assumptions, often at the cost of realism. Our contribution is to evaluate a Reinforcement Learning–based approach, motivated by the Markovian structure of the game described in Section 2, to compute the feedback Nash equilibria and to compare its performance with that of an iterative method.

RELATED WORK

In their overview of multi-agent reinforcement learning algorithms, Zhang, Yang, and Başar (7) highlight the difficulty of characterizing feedback Nash equilibria in differential games, especially in competitive settings, as classical value-based or policy-gradient methods often lack stability and robustness. To address continuous control problems of this kind, Lillicrap et al.(17) introduced the Deep Deterministic Policy Gradient (DDPG) algorithm, later improved by Fujimoto et al.(18) into Twin Delayed Deep Deterministic Policy Gradient (TD3) through clipped Double-Q learning. A related line of work extended DDPG to the distributional setting (D4PG) (9), which has been applied to pursuer–evader differential games (12), highlighting the potential of deep RL for dynamic strategic interactions. Building on this line, Lowe et al. (11) generalized DDPG to the multi-agent regime with a centralized critic and decentralized policies (MADDPG). Following this approach, we adapt TD3 to a multi-player configuration (MATD3) tailored to our setting.

## 2 PROBLEM SETUP

We consider a non-cooperative differential non-renewable resource extraction game with $N$ competing firms. Each firm starts with an initial stock $x_0 = 1$. The market price depends on the aggregate extraction rate, and all players act simultaneously under full information about the current stock levels.

At each time $t$, the system state is given by the vector $x(t) := (x_1(t), \ldots, x_N(t)) \in [0, x_0]^N$, representing all stock levels. With the extractions rates $\{q_i(t)\}_{1 \leq i \leq N}$ being fixed, the system evolves according to the following dynamic:

$$\dot{x}(t) = -q(t) := -(q_1(t), ..., q_N(t)). \tag{1}$$

Each player $i$ wants to maximize its long-term discounted profit:

$$J_{-i}(q_i(\cdot)) := \int_0^{+\infty} e^{-rt} F_i(x(t), q_1^*(t), \ldots, q_i(t), \ldots, q_N^*(t))\, dt$$

$$\text{subject to} \quad \dot{x}(t) = -\big(q_1^*(t), \ldots, q_i(t), \ldots, q_N^*(t)\big), \tag{2}$$

$$x(0) = (x_0, \ldots, x_0),$$

$$q_i(t) \in [0, 1].$$

where: $r = 5\%$ is the discount rate and $F_i$ is the profit function for player $i$. As previously introduced, we investigate two settings for the profit function:

**Physical depletion**

Extraction costs are neglected. Players extract until exhaustion–a situation often termed the "cake-eating" approach (6):

$$F_i\big(x(t), q(t)\big) = \left(1 - \sum_{j=1}^N q_j(t)\right) q_i(t). \tag{3}$$

This implies that, under both monopoly (13) and oligopoly (14), all stocks are depleted in finite time, depending on strategies. The game duration is thus defined as

$$T := \inf\left\{t \geq 0 : \forall i,\ x^{(i)}(t) = 0\right\} < +\infty. \tag{4}$$

**Economic depletion**

Each player faces a unit extraction cost that increases linearly with the remaining stock:

$$F_i\big(x(t), q(t)\big) = \left(c\, x_i(t) - \sum_{j=1}^N q_j(t)\right) q_i(t), \tag{5}$$

where $c > 0$ is the marginal cost parameter. This captures the rising economic difficulty of extraction as reserves decline, extending the game horizon to infinity (6).

In both cases, players aim to determine a strategy profile $(q_1^*, \ldots, q_N^*)$, conditioned on the current state of stocks, such that no player can improve their payoff by unilaterally deviating, given the strategies of the others. This defines a **Nash feedback equilibrium**.

## 3 LEARNING FRAMEWORK

As observed in the previous section, the structure of the game closely resembles the reinforcement learning settings, where agents explore an environment to learn strategies that maximize cumulative rewards. This analogy naturally motivates the use of an algorithm from the DDPG (17) family, due the continuous structure of our state-action spaces, as a scalable alternative to iterative methods for addressing the curse of dimensionality inherent in multi-player differential games.

### RL SETTINGS

To embed the game within an RL framework, we adopt a discrete-time formulation. Specifically, we introduce a uniform grid with step size $\delta t = 0.1$, yielding discrete times $t_k = k\,\delta t$ for $k \geq 0$. This discretization allows us to rewrite the objective function of player $i$ as :

$$\tilde{J}_{-i}\big(q_i(\cdot)\big) = \sum_{k=0}^{\infty} \gamma^k \, F_i\big(x(t_k), q_1^*(t_k), \ldots, q_i(t_k), \ldots, q_N^*(t_k)\big) \, \delta t, \tag{6}$$

where $\gamma = \frac{1}{1+r\,\delta t}$ denotes the discrete-time discount factor.

Within this discrete-time setup, the interaction naturally fits the standard RL paradigm: at each step $t_k$, the system is in state $x_{t_k}$, each agent $i$ selects an action $q_i(t_k)$ according to its policy $\pi_i$ so as to maximize its expected return, and receives an instantaneous reward $R_i(x_{t_k}, q(t_k))$.

This formulation further justifies our choice of the DDPG family, which relies on an actor–critic architecture. In this framework, the **Actor** is a deterministic policy network $\pi_\theta(x_t)$ mapping the current state to an action with the aim of maximizing the return estimated by the **Critic**. The Critic evaluates the expected cumulative reward of taking action $q_t$ in state $x_t$ through the value function $Q^\pi(x_t, q_t)$. The **Environment** then applies the action to the system dynamics and returns the next state and reward $(x_{t+\delta t}, R_t)$, while a **Replay Buffer** stores past transitions $(x_t, q_t, R_t, x_{t+1})$ along with a termination flag, enabling decorrelated sampling and stable off-policy learning.

To mitigate overestimation bias and improve stability, we adopt the TD3 variant (18), which extends DDPG with two independent Critics and uses the minimum of their estimates to reduce bias. In the single-agent case (Figure 1), this architecture reduces to a single Actor interacting with the environment under evaluation by the Critics, with the policy updated to maximize long-term profits subject to stock constraints (see Algorithm 1 in Appendix B). In the multi-agent case (Figure 2), we employ centralized training with Critics observing the joint state and actions, while each player is equipped with its own decentralized Actor mapping the global state to its extraction decision, making it the Multi-Agent TD3 (MATD3). This design captures strategic interdependencies during training while preserving independence at execution (see Algorithm 2 in Appendix C).

Finally, to prevent unrealistic over-extraction at the end of the game–where an agent could otherwise deplete its stock below zero–we enforce a physical constraint by introducing an effective extraction rate

$$q_{\text{eff}} = \min\big(q, \ \tfrac{x}{\delta t}\big), \tag{7}$$

which ensures that no more than the available quantity can be extracted within a single time step.

### STATE–ACTION SPACE DISCRETIZATION

To provide a credible benchmark for evaluating our reinforcement learning approach, we compare it against an iterative baseline grounded in classical value iteration (VI). Originally introduced by Bellman (19) in the simple control (single-agent) setting, value iteration is a dynamic programming scheme that solves the fixed-point equation associated with the Bellman operator $T$ applied to the value function $V(x) := \int_0^{+\infty} e^{-rt} F(x(t), q(t))dt$:

$$(TV)(x) = \max_{q \in \mathcal{Q}} \big\{ R(x,q) + \gamma\, V\big(x - q\,\delta t\big) \big\}, \tag{8}$$

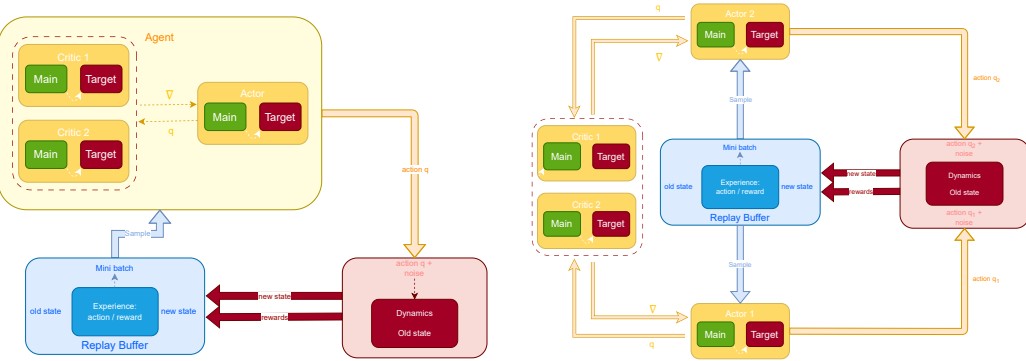

Figure 1: TD3 architecture in the single-agent case.

Figure 2: TD3 architecture in the two-player case.

on a fixed action grid (denoted $\mathcal{Q}$) for every point $x$ of the state grid (see Algorithm 3 in Appendix D).

In the multi-player setting, we adopt the algorithm of Cacace et al. (20), which extends value iteration to differential games by searching for fixed points of coupled Bellman operators. However, since the method may fail to converge to a configuration of mutual best responses (i.e., a local Nash equilibrium), we complement it with a Cournot tâtonnement step (see Algorithm 4 in Appendix E). This procedure iteratively adjusts the players' strategies according to their best-response mappings:

$$(q_1^*, q_2^*) \approx \left( \mathrm{Br}_1(q_2^*),\ \mathrm{Br}_2(q_1^*) \right), \tag{9}$$

where $\mathrm{Br}_i$ denotes the best-response operator of player $i$.

## 4 NUMERICAL EXPERIMENTS

We now present a series of numerical experiments, beginning with the single-agent case and progressively extending to duopoly and oligopoly settings, in order to evaluate the performance of our framework.

### SINGLE-AGENT PROBLEM

Since we have access to the theoretical solution of the single-agent problem under the physical depletion setting (see Appendix F), we are able to directly benchmark both approaches against it. As shown in Figure 3 and Figure 5 , the iterative method, implemented with a grid of 5000 stock points and 2500 action points, provides a closer fit to the theoretical trajectory, achieving an accuracy of 96.17%, compared to an average of 80.66% for the RL approach, evaluated over 50 Monte Carlo runs.

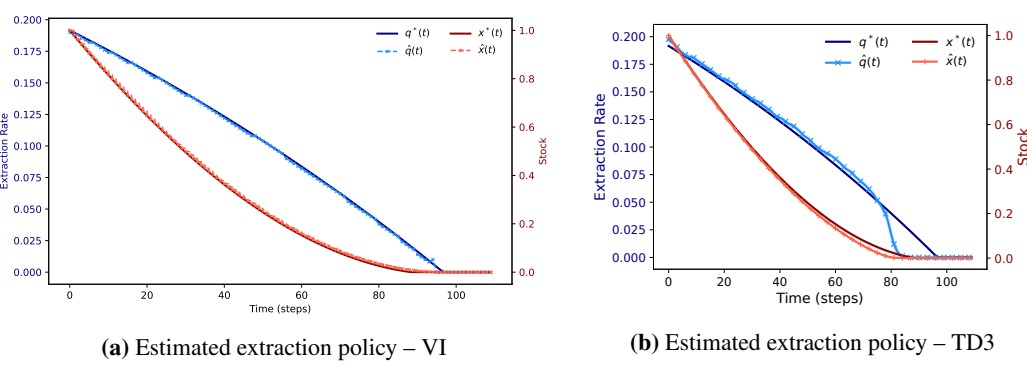

**(a)** Estimated extraction policy – VI

**(b)** Estimated extraction policy – TD3

Figure 3

In terms of total discounted reward, the iterative scheme attains 0.7350, while RL reaches 0.7352 on average, both being very close to the theoretical benchmark of 0.7371 (Figure 4). However, the RL agent exhibits a phenomenon of early stopping: on average, the game terminates after 81 iterations, whereas the iterative approach continues until 95 iterations, consistent with the theoretical horizon of 9.663 seconds.

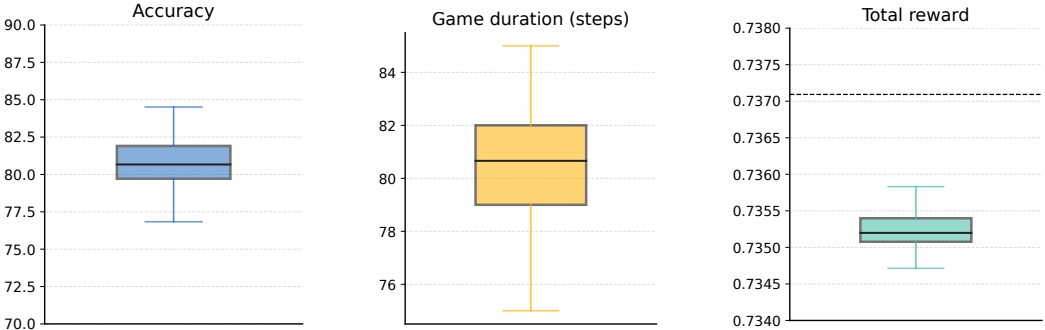

Figure 4: Metrics boxplots–single agent TD3

The RL framework remains consistent with the theoretical benchmark, as rewards converge toward but never reach the theoretical value as in Figure 6. And most importantly, the iterative method computes the solution within seconds, while RL requires about 20 minutes, as summarized in Table 1.

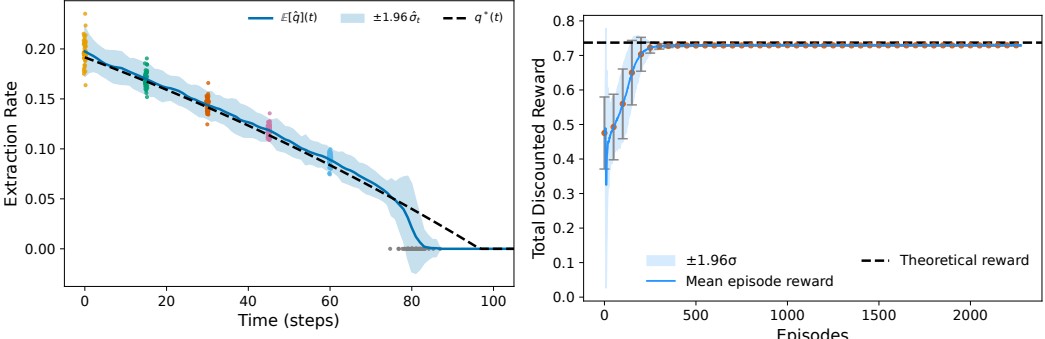

Figure 5: Estimated extraction policy for single agent with variability intervals

Figure 6: Learning curve for single agent

DUOPOLY PROBLEM

After examining the performance of RL in the monopoly (single-agent) case, we now turn to the competitive two-player setting. Since theoretical results are available under the economic depletion configuration (6) (See Appendix G), we perform a similar comparison to the single-agent case. In this case, the analytical solution converges asymptotically to zero, with extraction extending indefinitely. To allow comparison, we enforce termination at the step where the estimated solution falls below a tolerance of $10^{-3}$, using the same criterion for both methods.

With a coarse grid of $11 \times 11$ states and $51 \times 51$ actions, the iterative method achieved the results shown in Figure 7. The computation required 26 minutes and produced equilibrium rewards of $(0.1336, 0.1323)$, compared to the theoretical $(0.1340, 0.1340)$. Over 572 time steps, the corresponding accuracies were $(64.36\%, 72.41\%)$. Attempts to improve these values with a finer state grid of $51 \times 51$ led to execution times exceeding 6 hours, without yielding significant accuracy gains.

For the RL approach, the results were significantly better, with accuracies of $(79.74\%, 81.46\%)$ and average rewards of $(0.124, 0.124)$ (median rewards $(0.1307, 0.1303)$), as illustrated in Figure 8 (See Appendix I for the other metrics). The execution time remained stable between 35 and 40 minutes.

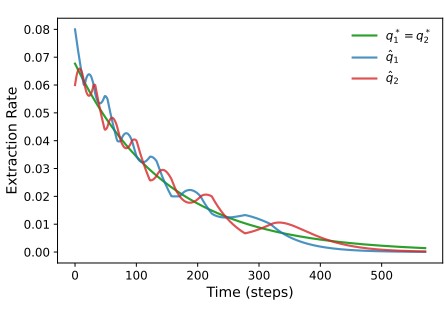 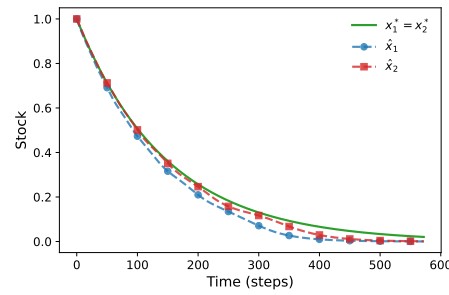

**(a)** Estimated extraction policy–MAVI

**(b)** Estimated stock evolution–MAVI

Figure 7: Duopoly economic depletion–MAVI

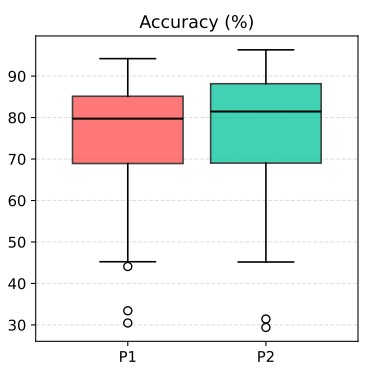 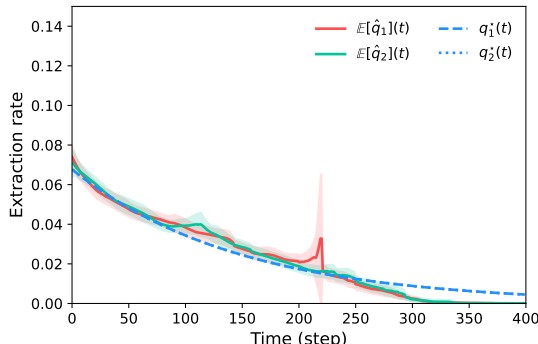

Figure 8: Dupoly economic depletion–MATD3

Similar observations hold for the physical depletion configuration, where both methods yield comparable rewards, but VI exhibits a markedly increasing execution time. A detailed summary of solutions and metrics is provided in Table 1 and in the Appendix H.

OLIGOPOLY PROBLEM

In this section, we get continue in the physical depletion configuration, and evaluate the performances for 3 players and 4 players configurations.

Figure 9 reports the 3-players results with MATD3, yielding average per-player rewards tightly clustered between $0.540$ and $0.544$. In comparison, MAVI achieved $(0.5625, 0.5626, 0.5624)$ per-player rewards, but with significantly different execution times: 35–40 minutes for MATD3 versus 102 minutes for MAVI using an $11 \times 11 \times 11$ state grid and a $51 \times 51 \times 51$ action grid.

Similar remarks hold for the 4-players case, where MATD3 runs in an average of 44 minutes, achieving rewards between $0.445$ and $0.450$, in contrast to more than four hours for MAVI with the same grids.

## 5 CONCLUSION

Our experiments highlight two main findings. First, the execution time of the iterative method increases drastically with the number of players, while the RL approach exhibits nearly stable runtimes across settings. Despite this difference in scalability, RL achieves rewards very close to those of the iterative method, with satisfactory accuracy against theoretical benchmarks.

Second, a limitation of RL is the phenomenon of early stopping, observed already in the single-agent case and amplified in duopoly and oligopoly, particularly under physical depletion. This arises partly from the imposed stopping rule at a tolerance threshold of $10^{-3}$, which prevents reaching the full

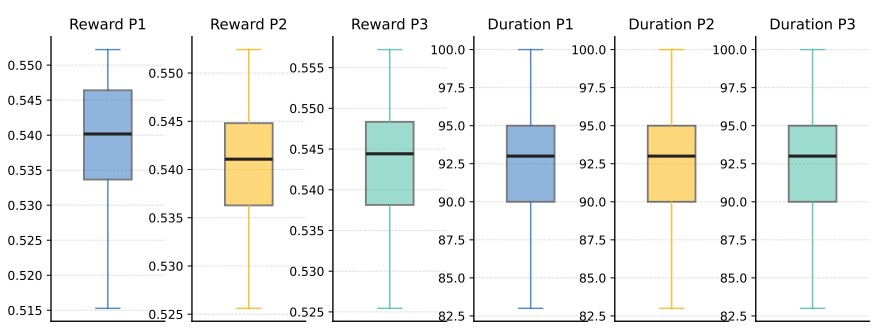

Figure 9: Metrics boxplots for 3 players–MATD3

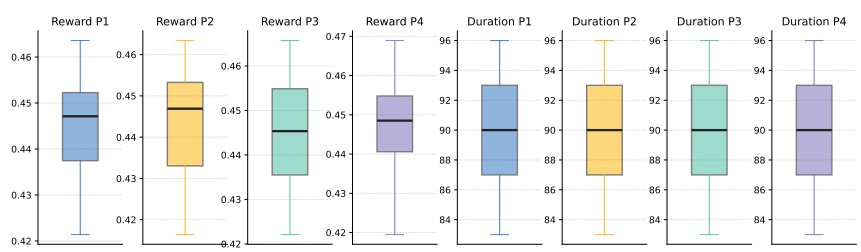

Figure 10: Metrics boxplots for 4 players–MATD3

theoretical game horizon. Additionally, agents tend to extract slightly more than the theoretical policy at each step, depleting their stock earlier: rather than collecting increasingly smaller rewards until the theoretical end of the game, they prefer harvesting the remaining stock in larger bursts near the end.

Table 1: Summary of performance and execution time for VI and TD3 across configurations.

| Configuration | Method | Reward (avg) | Accuracy (%) | Exec. Time |
|---|---|---|---|---|
| Single Agent (Physical.Dep) | VI | 0.7350 | 96.17 | 47.3 s |
| | TD3 | 0.7352 | 80.66 | 20 min |
| Duopoly (Economic.Dep) | MAVI | (0.1336, 0.1323) | (64.36, 72.41) | 34 min |
| | MATD3 | (0.124, 0.124) | (79.74, 81.46) | 35–40 min |
| Duopoly (Physical.Dep) | MAVI | (0.630,0.628) | – | 26 min |
| | MATD3 | (0.634,0.634) | – | 35–40 min |
| Oligopoly (3 players) | MAVI | (0.5625, 0.5626, 0.5624) | – | 102 min |
| | MATD3 | (0.540,0.541,0.544) | – | 35–40 min |
| Oligopoly (4 players) | MAVI | 0.454 | – | > 4 h |
| | MATD3 | (0.448,0.448,0.445,0.449) | – | 40-45 min |

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

## A    PROBLEM TAXONOMY IN RESOURCE GAMES

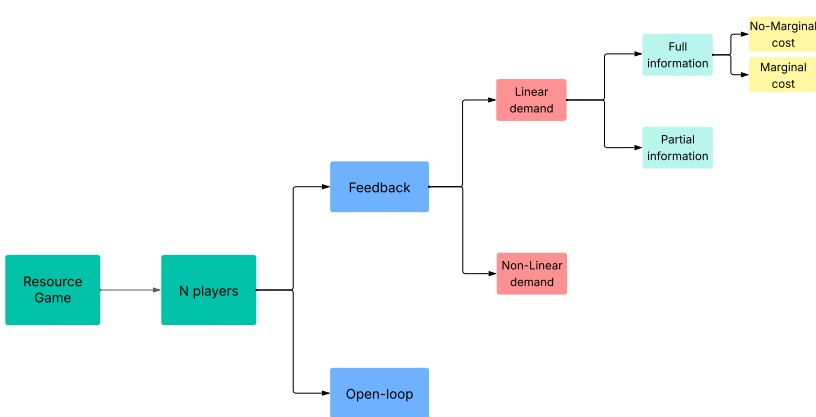

Figure 11: Problem Taxonomy in Resource Games

## B    TD3 FOR SINGLE-AGENT

---

**Algorithm 1:** TD3 for single-agent

---

**Input:** $\gamma, \tau, \sigma, c, d, \delta t$, batch size $B$, episodes $M$, horizon $T$

**Output:** Policy $\pi_\theta$

1   Init $Q_{\phi_1}, Q_{\phi_2}, \pi_\theta$; targets $\phi'_j \leftarrow \phi_j$ $(j = 1, 2)$, $\theta' \leftarrow \theta$; buffer $\mathcal{B}$;

2   **for** $m = 1$ **to** $M$ **do**

3     $x \leftarrow x_0$;

4     **for** $t = 1$ **to** $T$ **do**

5       $q \leftarrow \mathrm{clip}(\pi_\theta(x) + \epsilon, 0, 1), \epsilon \sim \mathcal{N}(0, \sigma^2)$;

6       $q_{\mathrm{eff}} \leftarrow \min(q, x/\delta t)$;

7       $(x', R, \mathrm{done}) \leftarrow \mathrm{EnvStep}(x, q_{\mathrm{eff}})$; store $(x, q, R, x', \mathrm{done})$; $x \leftarrow x'$;

8       **if** done **then**

9       **end**

10      sample $\{(x_i, q_i, R_i, x'_i, \mathrm{done}_i)\}_{i=1}^{B}$;

11      $\tilde{q}_i \leftarrow \pi_{\theta'}(x'_i) + \mathrm{clip}(\epsilon_i, -c, c), \epsilon_i \sim \mathcal{N}(0, \sigma^2)$;

12      $y_i \leftarrow R_i + \gamma(1 - \mathrm{done}_i) \min_j Q_{\phi'_j}(x'_i, \tilde{q}_i)$;

13      $\phi_j \leftarrow \arg\min_{\phi_j} \frac{1}{B} \sum_i (Q_{\phi_j}(x_i, q_i) - y_i)^2, \; j = 1, 2$;

14      **if** $t \bmod d = 0$ **then**

15        $\theta \leftarrow \arg\max_\theta \frac{1}{B} \sum_i Q_{\phi_1}(x_i, \pi_\theta(x_i))$;

16        $\phi'_j \leftarrow \tau\phi_j + (1 - \tau)\phi'_j, j = 1, 2; \; \theta' \leftarrow \tau\theta + (1 - \tau)\theta'$;

17      **end**

18    **end**

19 **end**

---

## C  TD3 FOR MULTI-AGENT (MATD3): 2 PLAYERS

---

**Algorithm 2:** TD3 for two agents

---

**Input:** $\gamma, \tau, \sigma, c, d, \delta t$, batch size $B$, episodes $M$, horizon $T$

**Output:** Policies $\pi_{\theta_1}^{(1)}, \pi_{\theta_2}^{(2)}$

1 Init $Q_{\phi_1}, Q_{\phi_2}$ (output $(Q_1, Q_2)$), $\pi_{\theta_1}^{(1)}, \pi_{\theta_2}^{(2)}$; targets $\phi_j' \leftarrow \phi_j, \theta_k' \leftarrow \theta_k$ ($j, k = 1, 2$); buffer $\mathcal{B}$;

2 **for** $m = 1$ **to** $M$ **do**

3      $x \leftarrow x_0$;

4      **for** $t = 1$ **to** $T$ **do**

5          $q \leftarrow \mathrm{clip}([\pi_{\theta_1}^{(1)}(x), \pi_{\theta_2}^{(2)}(x)] + \epsilon, 0, 1)$, $\epsilon \sim \mathcal{N}(0, \sigma^2 I)$;

6          $q_{\mathrm{eff}} \leftarrow \min(q, x/\delta t)$;

7          $(x', R, \mathrm{done}) \leftarrow \mathrm{EnvStep}(x, q_{\mathrm{eff}})$; store $(x, q, R, x', \mathrm{done})$; $x \leftarrow x'$;

8          **if** done **then**

9          **end**

10          sample $\{(x_i, q_i, R_i, x_i', \mathrm{done}_i)\}_{i=1}^{B}$;

11          $\tilde{q}_i \leftarrow [\pi_{\theta_1'}^{(1)}(x_i'), \pi_{\theta_2'}^{(2)}(x_i')] + \mathrm{clip}(\epsilon_i, -c, c)$, $\epsilon_i \sim \mathcal{N}(0, \sigma^2 I)$;

12          $y_{i,k} \leftarrow R_{i,k} + \gamma(1 - \mathrm{done}_i) \min_j (Q_{\phi_j'}(x_i', \tilde{q}_i))_k$,   $k = 1, 2$;

13          $\phi_j \leftarrow \arg\min_{\phi_j} \frac{1}{B} \sum_i \sum_{k=1}^{2} \left((Q_{\phi_j}(x_i, q_i))_k - y_{i,k}\right)^2$, $j = 1, 2$;

14          **if** $t \bmod d = 0$ **then**

15              $\theta_1 \leftarrow \arg\max_{\theta_1} \frac{1}{B} \sum_i (Q_{\phi_1}(x_i, [\pi_{\theta_1}^{(1)}(x_i), \pi_{\theta_2}^{(2)}(x_i)]))_1$;

16              $\theta_2 \leftarrow \arg\max_{\theta_2} \frac{1}{B} \sum_i (Q_{\phi_1}(x_i, [\pi_{\theta_1}^{(1)}(x_i), \pi_{\theta_2}^{(2)}(x_i)]))_2$;

17              $\phi_j' \leftarrow \tau\phi_j + (1 - \tau)\phi_j'$, $\theta_k' \leftarrow \tau\theta_k + (1 - \tau)\theta_k'$;

18          **end**

19      **end**

20 **end**

---

## D  VALUE ITERATION FOR SINGLE AGENT

---

**Algorithm 3:** VI for single agent

---

**Input:** $\gamma, \delta t$, grids $\mathbf{X} \subset [0,1], \mathcal{Q} \subset [0,1]$, tolerances $\varepsilon_{\mathrm{vi}}, \varepsilon_{\mathrm{term}}$

**Output:** Approximate value $V$ and greedy policy $q^*$

1 *Let* $\Pi_{\leftarrow}(x)$ be the nearest point in $\mathbf{X} \le x$ (and 0 if $x < 0$).;

2 Initialize $V^{(0)}(x_i) \leftarrow 0$ for all $x_i \in \mathbf{X}$; $n \leftarrow 0$;

3 **repeat**

4      **foreach** $x_i \in \mathbf{X}$ **do**

5          **if** $x_i < \varepsilon_{\mathrm{term}}$ **then** $V^{(n+1)}(x_i) \leftarrow 0$;

6          **else** $V^{(n+1)}(x_i) \leftarrow$

             $\max_{q \in \mathcal{Q}} \left[ R\big(x_i, \underbrace{\min\{q, \, x_i/\delta t\}}_{q_{\mathrm{eff}}}\big) + \gamma V^{(n)}\big(\Pi_{\leftarrow}(x_i - \underbrace{\min\{q, \, x_i/\delta t\}}_{q_{\mathrm{eff}}} \cdot \delta t)\big) \right]$;

7          ;

8      **end**

9      $n \leftarrow n + 1$;

10 **until** $\|V^{(n+1)} - V^{(n)}\|_\infty < \varepsilon_{\mathrm{vi}}$;

11 **return** $V^{(n)}$ and

     $q^*(x_i) = \arg\max_{q \in \mathcal{Q}} \left[ R\big(x_i, \underbrace{\min\{q, \, x_i/\delta t\}}_{q_{\mathrm{eff}}}\big) + \gamma V^{(n)}\big(\Pi_{\leftarrow}(x_i - \underbrace{\min\{q, \, x_i/\delta t\}}_{q_{\mathrm{eff}}} \cdot \delta t)\big) \right]$;

---

## E  VALUE ITERATION FOR MULTI-AGENT (MAVI): 2 PLAYERS

---

**Algorithm 4:** VI for two agents

---

**Input:** $\delta t > 0$, interest rate $r$, discrete sets $Q_1^{\#}, Q_2^{\#}$, tolerances $\varepsilon_1, \varepsilon_2$
**Output:** Approximate values $V_1, V_2$ and policies $q_1^*, q_2^*$

1   Initialize $V_1^{(0)}, V_2^{(0)}$; $k \leftarrow 0$;
2   **repeat**
3      **for** $j = 1$ **to** $N$ **do**
4         $q_1^* \leftarrow \arg\min_{q_1 \in Q_1^{\#}} \left\{ \frac{1}{1+r\delta t} V_1^{(k)}(\text{next}(x^j, q_1, q_2^*)) + \frac{\delta t}{1+r\delta t} R_1(x^j, q_1, q_2^*) \right\} :=$
         $Br_1(q_2^*)$;
5         $q_2^* \leftarrow \arg\min_{q_2 \in Q_2^{\#}} \left\{ \frac{1}{1+r\delta t} V_2^{(k)}(\text{next}(x^j, q_1^*, q_2)) + \frac{\delta t}{1+r\delta t} R_2(x^j, q_1^*, q_2) \right\} :=$
         $Br_2(q_1^*)$;
6         **if** $(q_1^*, q_2^*)$ *not found* **then**
7            **for** $t = 1$ **to** $8$ **do**
8              $q_1^{t+1} \leftarrow Br_1(q_2^t)$;
9              $q_2^{t+1} \leftarrow Br_2(q_1^{t+1})$;
10            **end**
11            $q_1^*, q_2^* \leftarrow q_1^{t+1}, q_2^{t+1}$;
12         **end**
13         $V_i^{(k+1)}(x^j) \leftarrow \frac{1}{1+r\delta t} V_i^{(k)}(\text{next}(x^j, q^*)) + \frac{\delta t}{1+r\delta t} R_i(x^j, q^*), \quad i = 1, 2$;
14      **end**
15      $k \leftarrow k + 1$;
16   **until** $\|V_i^{(k+1)} - V_i^{(k)}\| < \varepsilon_i, \ i = 1, 2$;

---

## F  MONOPOLY PHYSICAL DEPLETION: THEORETICAL RESULTS

In this situation, the problem simplifies to the following optimal control problem:

$$\text{Maximize} \quad J(q(\cdot)) := \int_0^{+\infty} e^{-rt}\, [1 - q(t)]\, q(t)\, dt.$$
$$\text{subject to} \quad \dot{x}(t) = -q(t)$$
$$x(0) = x_0$$
$$q(t) \in [0, 1]$$

By definition of the stopping time $T$, we know that: $x(T) = 0$, meaning that : $x_0 = \int_0^T q(t)\, dt$.

In addition, the Hamiltonian function of this problem can be written as:

$$H(x, q, \lambda, t) = (1 - q)q - \lambda q. \tag{10}$$

According to the PMP, the optimality conditions are:

$$(\forall t \leq T): \quad \begin{cases} \dfrac{\partial H}{\partial q}(x, q^*, \lambda, t) = 0. \\[2mm] \lambda'(t) = r\, \lambda(t) - \dfrac{\partial H}{\partial x}(x, q, \lambda, t). \end{cases} \tag{11}$$

This leads to a first relation between the optimal strategy and the costate:

$$q^*(t) = \frac{1 - \lambda(t)}{2} \quad \text{and} \quad \lambda'(t) = r\lambda(t) \tag{12}$$

$$\Rightarrow q^*(t) = \frac{1 - \lambda_0 e^{rt}}{2} \quad \text{and} \quad \lambda(t) = \lambda_0 e^{rt} \tag{13}$$

In order to determine the constants $\lambda_0$ and $T$, we use the fact that : $q^*(T) = 1 - \lambda_0 e^{rT} = 0$ and $x(T) - x_0 = -\int_0^T q^*(t)dt$.

Meaning that $\lambda_0 = e^{-rT}$ and :

$$x_0 = \int_0^T q^*(t)dt$$

$$\Rightarrow x_0 = \int_0^T \frac{1 - \lambda_0 e^{rt}}{2} dt$$

$$\Rightarrow x_0 = \int_0^T \frac{1 - e^{r(t-T)}}{2} dt$$

$$\Rightarrow x_0 = \frac{T}{2} + \frac{1}{2r}(e^{-rT} - 1)$$

Numerical solving gives: $\mathbf{T = 9.663}$ and $\boldsymbol{\lambda_0 = 0.616}$.

## G DUOPOLY ECONOMIC DEPLETION: THEORITICAL RESULTS

As described in (6), the linear cost structure introduced allows for a feedback equilibrium in which the optimal extraction rates are linear functions of the stock levels:

$$q_1^* = \alpha_1 x_1 + \alpha_2 x_2, \quad q_2^* = \mu_1 x_1 + \mu_2 x_2. \tag{14}$$

Due to the symmetric configuration we consider (identical marginal costs and discount rates), the parameters are also symmetric: $\alpha_1 = \mu_2$ and $\alpha_2 = \mu_1$. They can be derived using the following formulas:

$$\mu_1 = -a_1(1 + s^2)^{-1/2}, \quad \mu_2 = (a_2 s + \tfrac{1}{2}a_1)(1 + s^2)^{-1/2} - \tfrac{1}{2}r. \tag{15}$$

Where :

$$s = \frac{a_0}{3}\left\{p_0 + 2p_1 \cos\left[\tfrac{1}{3}\arccos\left(\frac{p_2}{p_1^3}\right)\right]\right\},$$

$$p_0 = 3\gamma + 5, \quad p_1 = \sqrt{64 + 60\gamma + 9\gamma^2}, \quad p_2 = 404 + 666\gamma + 270\gamma^2 + 27\gamma^3,$$

$$a_0 = \frac{1}{\sqrt{7}}, \quad a_1 = \sqrt{\frac{r(2c+r)}{7}}, \quad a_2 = \frac{1}{2}\sqrt{r(2c+r)}, \quad \gamma = \frac{r}{c}.$$

Numerically it gives: $\boldsymbol{\mu_1 = -0.029}$ and $\boldsymbol{\mu_2 = 0.097}$.

## H DUOPOLY ECONOMIC DEPLETION: TD3 METRICS

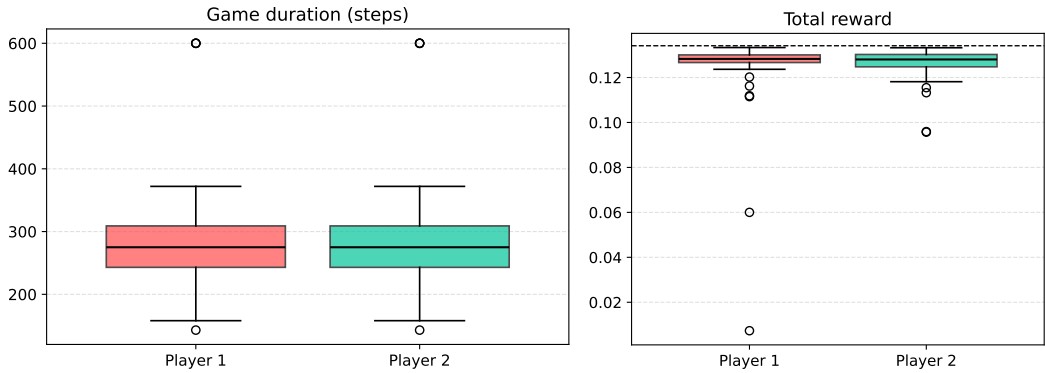

Figure 12: Metrics boxplots for duopoly economic depletion.

## I DUOPOLY PHYSICAL DEPLETION

We present here the extraction policies, stock evolutions, and key metrics estimated by both methods in the duopoly physical depletion setting.

For MAVI, the obtained reward was $(0.630, 0.628)$ over 162 time steps (Figure 13), with an execution time of 26 minutes.

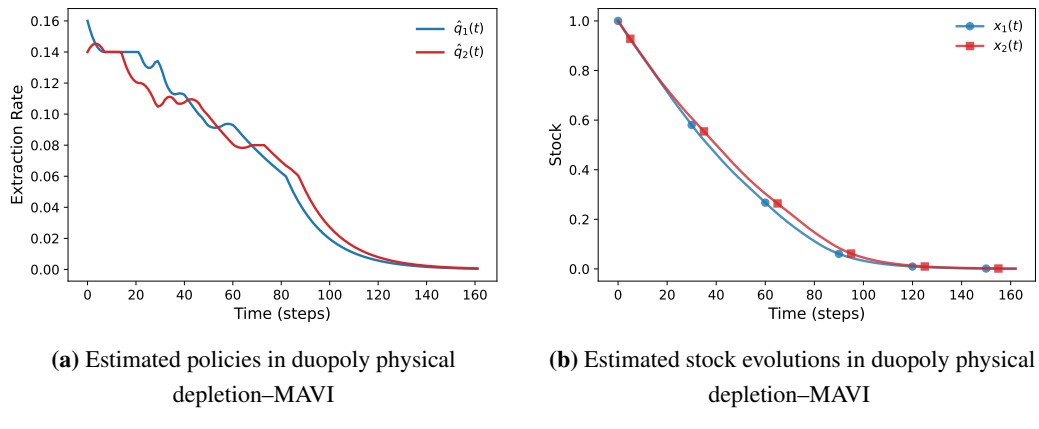

**(a)** Estimated policies in duopoly physical depletion–MAVI

**(b)** Estimated stock evolutions in duopoly physical depletion–MAVI

Figure 13

As for MATD3, each player achieved a reward of $(0.634, 0.634)$ over 94 time steps (Figure 14), with an execution time of 35–40 minutes.

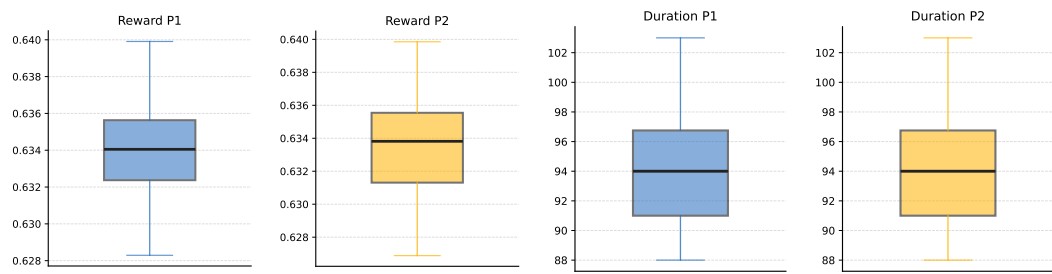

Figure 14: Metrics boxplots for duopoly physical depletion–MATD3

Regarding the estimated extraction policies, the early-stopping phenomenon is once again observed (Figure 15).

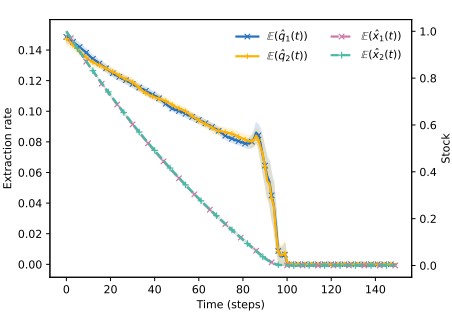

**(a)** Estimated policies and stocks in duopoly physical depletion–MATD3

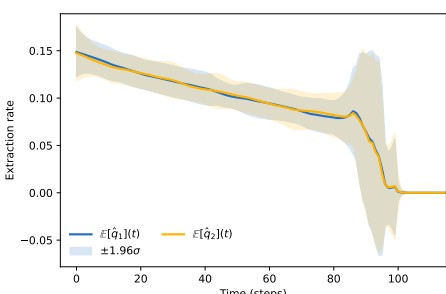

**(b)** Variability intervals in duopoly physical depletion–MATD3

Figure 15

