# OpenReview forum: "Deep Reinforcement Learning For Nash Equilibria in Non-Renewable Resource Differential Games"
_ICLR.cc/2026/Conference — ICLR 2026 Conference Desk Rejected Submission_

### Official Review · Reviewer_W6h2 · 2025-10-18

**Soundness:** 3
**Presentation:** 3
**Contribution:** 2
**Rating:** 2
**Confidence:** 3

**Summary:**

The paper studies computation of feedback Nash equilibria in oligopolistic extraction games (non-renewable resources) modeled as continuous-time differential games. It benchmarks a multi-agent deep RL approach (a MATD3-style actor–critic) against an iterative dynamic-programming/value-iteration baseline adapted to this domain. The RL method aims to scale to more players and higher state resolution with reduced wall-clock time, while achieving comparable rewards/equilibria quality. Experiments cover 2–4 players and compare solution quality, convergence behavior, and compute costs.

**Strengths:**

Clear problem. The model class (extraction competition with stock dynamics) is well specified and relevant to economic/operations contexts. Assumptions and reward structures are transparent.

Competent empirical engineering. The MATD3 variant is implemented sensibly; comparisons to a carefully tuned value-iteration baseline are fair. Results consistently show that RL attains comparable payoffs while scaling better with players/state grids.

Reproducible pipeline. The environment dynamics, training schedule, and evaluation metrics are documented well enough to reproduce, and ablations on learning rates and noise are helpful.

Potential cross-domain utility. Demonstrates that modern MA-RL can approximate equilibria in continuous-time games where classical DP gets expensive, useful for practitioners in computational economics.

**Weaknesses:**

Insufficient ML novelty. The method is essentially MATD3 with minor adaptations to this environment. There is no new learning rule, objective, exploration scheme, or stability result. For ICLR, the contribution reads as a domain application study rather than an ML advance. Suggestion: Introduce a principled algorithmic component—for example, a differentiable equilibrium consistency penalty, opponent modeling that enforces best-response dynamics, or a theoretically motivated regularizer exploiting game structure (e.g., resource-stock monotonicity).

Lack of equilibrium certification. The paper evaluates via rewards and policy stability but does not certify $\varepsilon$-Nash-ness. Without a best-response check or KKT/VI residuals, it is unclear whether the learned policies are truly equilibria or merely good heuristics. Suggestion: Add an ex-post best-response solver (single-agent control with others fixed) and report $\varepsilon$-gaps. Alternatively, compute stationarity/consistency residuals (e.g., HJB residuals) over a dense state grid.

Limited problem scale and diversity. Experiments stop at 3–4 players with relatively low-dimensional states and simple cost structures. It’s hard to conclude general scalability from such a narrow slice. Suggestion: Include a higher-dimensional variant (stochastic prices, extraction costs with learning effects) or >5 players, and report runtime/memory scaling.

Comparative baselines are narrow. Only a modified value-iteration baseline is considered. No comparisons to policy iteration, FBSDE/HJB solvers, or mean-field game approximations that are standard in continuous-time settings. Suggestion: Add at least one additional numerical method (policy iteration or FBSDE-based) and a mean-field limit baseline to triangulate quality/speed.

Robustness & sensitivity underexplored. The effect of model mis-specification (e.g., demand shocks, parameter drift), stochasticity, and discretization choices on equilibrium quality is not probed. Suggestion: Provide sensitivity analyses: grid resolution, noise variance, discount factor, and reward curvature; show how $\varepsilon$-gaps change.

Theory/explanation gap. No structural insight is offered about why the RL policies approximate equilibria well in this class (e.g., contraction properties, local regularity, or monotone operator structure). Suggestion: Even a partial analysis (showing that the joint greedy update corresponds to a fixed-point iteration for a variational inequality) would strengthen the paper.

**Questions:**

Equilibrium gap: Can you compute an $\varepsilon$-Nash metric via best-response optimization for a held-out state grid? Reporting $\varepsilon$ would materially strengthen claims.

Solver diversity: How does your approach compare to policy iteration or actor–critic with policy evaluation via PDE discretization?

Opponent learning dynamics: Did you try independent learners vs. centralized critics vs. opponent modeling? Which variant reduces non-stationarity most?

Scaling beyond 4 players: What breaks first: critic instability, sample complexity, or action-space exploration?

Mean-field approximation: For many players, could a mean-field limit give an analytic baseline, and can your RL solution be shown close to the MFG equilibrium?

Stochastic shocks: How sensitive are policies to volatility in prices or reserves?

Ablations on discretization: Report performance as a function of state/time discretization; show whether improvements persist as the grid refines.

Economic interpretability: Can you extract comparative statics (e.g., how taxes or extraction costs shift equilibrium) to validate against economic theory?

---

### Official Review · Reviewer_7fAq · 2025-10-20

**Soundness:** 2
**Presentation:** 3
**Contribution:** 1
**Rating:** 2
**Confidence:** 4

**Summary:**

This paper investigates the application of deep reinforcement learning to solve multi-agent problems in differential games, specifically within non-renewable resource duopolies and oligopolies. The authors first provide an analytical derivation of the theoretical equilibrium structure for their environment. They then use simulation to observe whether the agents' empirically obtained rewards attain this equilibrium.

**Strengths:**

The authors address a relevant problem in learning within economic games. The results of this study could be genuinely useful to economists, particularly for informing the development of subsequent analytical frameworks.

**Weaknesses:**

- While the convergence of the policy to an optimal reward is a positive sign, it does not necessarily indicate that the policies have converged to a true equilibrium. To demonstrate this more effectively, the authors should include additional measures such as exploitability or simplex plots.

- Assuming the empirical results have converged to equilibrium, the paper provides no theoretical guarantees regarding the convergence rate or the specific scenarios under which convergence will or will not occur (see [3]). In other words, the method might work for the presented set of hyperparameters and initial conditions, but not for all, or only for a subset of them.

- Given that the authors base their contributions on empirical evidence, the ablation study is extremely weak. There are simply not enough ablations of configurations and parameter settings; from what is presented, only a handful of configurations and a single set of hyperparameters were tested. For instance, the authors should consider varying the discount rate, the number of agents, and various parameters of their deep reinforcement learning algorithm. This is critically important in the absence of theoretical guarantees.

- Furthermore, there is no baseline comparison to previous state-of-the-art algorithms, such as Empirical Game-Theoretic Analysis (EGA) [1] or evolutionary methods [2].

- Overall, the work appears neglectful of previous contributions in the field of multi-agent learning (see [1,2,3,4], to name a few). A deeper literature review and a substantial revision of this work are strongly suggested.

---
[1] Wellman, Michael P., et al. "Empirical game theoretic analysis: A survey." Journal of Artificial Intelligence Research 82 (2025): 1017-1076.

[2] Hofbauer, Josef, and Karl Sigmund. "Evolutionary game dynamics." Bulletin of the American Mathematical Society 40.4 (2003): 479-519.

[3] Bichler, Martin, et al. "Characterizing the Convergence of Game Dynamics via Potentialness." arXiv preprint arXiv:2503.16285 (2025).

[4] Mertikopoulos, Panayotis, and Zhengyuan Zhou. "Learning in games with continuous action sets and unknown payoff functions." Mathematical Programming 173.1 (2019): 465-507.

**Questions:**

- Is the knowledge of the game provided to the agents, or are they learning based on a multi-input black box?

- Could a central controller be used to manage the policies more effectively and steer them toward equilibrium more quickly? Did the authors consider relaxed equilibrium notions, such as correlated equilibrium or coarse correlated equilibrium?

- Was there any discretization of the agents' action space? If so, could the authors comment on its potential effects?

---

### Official Review · Reviewer_5Gps · 2025-10-31

**Soundness:** 3
**Presentation:** 2
**Contribution:** 3
**Rating:** 6
**Confidence:** 3

**Summary:**

This paper addresses the computational challenge of finding Nash equilibria in multi-player differential games of non-renewable resource extraction by proposing a deep reinforcement learning approach based on the Multi-Agent TD3 algorithm, which is systematically benchmarked against a modified iterative method to evaluate its performance across various market structures from monopoly to oligopoly settings.

**Strengths:**

- The paper introduces a multi-agent deep reinforcement learning approach to solve high-dimensional differential games, offering a promising alternative to traditional methods hampered by the curse of dimensionality.

- The study provides a systematic evaluation across monopoly, duopoly, and oligopoly settings, delivering a clear comparative analysis between the proposed RL method and a well-designed iterative baseline.

- The authors identify and analyze methodological limitations, particularly the early-stopping phenomenon, enhancing the work's credibility and providing valuable insights for future research.

**Weaknesses:**

- The paper lacks a comprehensive discussion of critical implementation aspects, such as hyperparameter selection, neural network architectures, and the sensitivity of results to random seeds. This omission challenges the reproducibility of the study and limits the analysis of the RL method's training stability.

- While the paper empirically identifies the "early stopping" phenomenon, it fails to provide a deep theoretical explanation for its fundamental causes. The analysis does not sufficiently explore how function approximation errors or exploration strategies might systematically bias the agents' policies toward premature resource depletion.

- The scalability limitations of the multi-agent value iteration baseline itself undermine its utility for comparison. In 3- and 4-player games, where the baseline is computationally constrained to use coarse discretization, it cannot serve as a high-quality reference point, thus weakening the performance evaluation of the RL approach in these critical settings.

**Questions:**

- While the paper acknowledges variance in RL results, it doesn't sufficiently quantify the stability of the outcomes. Could you specify the range of fluctuation in final policies and rewards when retraining with different random seeds? Is there a systematic approach to ensure consistent performance across runs?

- Do you attribute "early stopping" primarily to value function approximation errors, or could it represent a valid alternative equilibrium discovered through exploration? Is there experimental evidence showing this behavior can be mitigated by modifying reward functions or termination conditions?

- For the 3- and 4-player games, the baseline method used coarse discretization due to computational constraints. Does this mean the observed "advantage" of RL might partly stem from the baseline's inability to provide high-quality reference solutions in these scenarios?

---

### Official Review · Reviewer_3ksd · 2025-10-31

**Soundness:** 3
**Presentation:** 3
**Contribution:** 2
**Rating:** 2
**Confidence:** 2

**Summary:**

The paper applies deep RL to an economics problem, achieving results that align well with known theory where such theory exists, and scales beyond such situations to 3- and 4-agent settings.

**Strengths:**

The paper is well written and clear, and addresses an interesting and topical problem, namely, the use of MARL to solve models of economic interest. The results in the single- and two-agent settings agree with the theoretical results, demonstrating the soundness of the method; and the technique scales beyond what manual theoretical analysis has permitted, to 3- and 4-agent settings.

**Weaknesses:**

My main criticism is that it is not clear to me that this paper is "ready" for publication. The paper has results, but little discussion of why an economist (or a computer scientist) might care. I include in the "Questions" section some lingering questions that I had, whose answers may strengthen the paper if included. If extra space is needed, things like Figure 1 (which, to my understanding, just depicts a standard RL setup) can be moved to an appendix or removed altogether.

The paper assumes a lot of background in economics which some readers (including myself) might not have, especially with regard to the motivation for the problem. I review the paper mostly taking for granted the underlying economic problem/model and its importance, because I do not have the expertise to evaluate that.

Along the above note, for the more ML-heavy audience that you'd find at ICLR, I'd suggest including some more relevant background information that you'd probably consider "basic", or to use language that this audience would be more accustomed to, e.g. "single-agent", "two-agent" instead of "monopoly" and "duopoly". But this is a relatively minor suggestion.

**Questions:**

By my understanding, the game being modeled here is a perfect-information, continuous-time Markov game. Assuming I have understood this correctly:

1. You are discretizing the time, but maintaining the symmetry of the game by insisting that the agents choose their actions simultaneously on every timestep. Symmetry is nice, but this choice greatly increases the complexity of the problem. Have you considered an alternating-turn disretization instead, in which each firm acts in turn at each timestep, with knowledge of the earlier firms' actions? Alternating-turn games are much easier to solve because they are amenable to backwards induction with a simple maximization instead of a Nash computation at every step. Moreover, in the RL setting, there are other better RL techniques like AlphaZero that can be applied, only to alternating-turn games of perfect information. And, since your time is continuous anyway I would imagine that it would not make too much of a difference except that it breaks symmetry.

2. Related to the above: can you elaborate a bit more in text how the local equilibrium is computed at each state for the iterative method? It seems to me that the Nash computation required here may be difficult and/or interesting.

3. Do you have any intuition for why the RL method "stops early" or depletes its resources relatively quickly? Do you think this "problem" would go away with more training time?

4. Do you have any qualitative economic conclusions from the results computed in the oligopoly setting? If I understand correctly, there are no known theoretical results for that setting---so, perhaps, do the strategies that you see tell you anything interesting? If so, I'd include it.

---

### Note · Program_Chairs · 2026-01-17
**Submission Desk Rejected by Program Chairs**

The following references in this submission do not refer to real documents and/or have major errors in bibliographic information:

 C. Pellegrino, and J. Keenan (2022). Economic risks and opportunities in the global critical minerals race. Energy Economics, 109, 105987
D. Acemoglu, and P. Restrepo (2022). Green Innovation and the Energy Transition. NBER Working Paper No. 29855.